# The Influence of Home-Based Music Therapy Interventions on Relationship Quality in Couples Living with Dementia—An Adapted Convergent Mixed Methods Study

**DOI:** 10.3390/ijerph20042863

**Published:** 2023-02-06

**Authors:** Kristi Stedje, Tone Sæther Kvamme, Kjersti Johansson, Tanara Vieira Sousa, Helen Odell-Miller, Karette Annie Stensæth, Anna A. Bukowska, Jeanette Tamplin, Thomas Wosch, Felicity Anne Baker

**Affiliations:** 1Centre for Research in Music and Health, Norwegian Academy of Music, 0369 Oslo, Norway; 2Creative Arts and Music Therapy Research Unit, The University of Melbourne, Melbourne, VIC 3010, Australia; 3Cambridge Institute for Music Therapy Research, Anglia Ruskin University, Cambridge CB1 1PT, UK; 4Institute of Applied Sciences, University of Physical Education in Krakow, 31-571 Krakow, Poland; 5Music Therapy Lab, Institute for Applied Social Sciences, University of Applied Sciences Würzburg-Schweinfurt, 97072 Würzburg, Germany

**Keywords:** music therapy, music therapy interventions, dementia, relationship quality, couples, couplehood

## Abstract

Relationship quality is important for well-being and quality of life in couples living with dementia. Home-based music therapy interventions may be conducted with the aim of enhancing relationship quality. However, the effects or influences of such interventions are only briefly investigated in previous studies. This study’s aim was to identify how a 12-week home-based music therapy intervention may influence relationship quality in couples living with dementia, through an adapted convergent mixed methods design. In this case, 68 participating couples from the HOMESIDE RCT study, and four individually recruited couples, received the music therapy intervention. Relationship quality for all participants was measured by the standardized Quality of Caregiver-Patient Relationship scale, and qualitative interviews were conducted with the four individually recruited participants at baseline and post intervention. Quantitative analysis indicated no statistically significant intervention effect. However, relationship quality remained stable over the intervention period. The qualitative analysis identified that the music therapy interventions primarily led to positive emotions, closeness, intimacy, and communication between the persons with dementia and their care partners. Intervention influences could also be ambiguous, as sharing music experiences might involve a risk of evoking vulnerabilities or negative emotional responses.

## 1. Introduction

About 50 million people live with dementia globally, and the number is continuously rising [1]. People with dementia are in need of high-quality care interventions that are person-centred and include family members and others close to the person with dementia [1]. Relationship quality in couples living with dementia is a significant factor impacting well-being and quality of life for the person with dementia and their partner [2], and being in intimate relationships is defined as an important need for both persons with dementia living at home and their caregivers [3]. Life experiences associated with a dementia diagnosis may influence relationship quality in married or co-habiting couples [4]. As the symptoms of dementia worsen with disease progression, several factors may contribute positively or negatively to changes in relationship quality. Such factors may be multifaceted and complex and vary from couple to couple. Changes in functioning caused by cognitive decline or behavioural and psychological symptoms of dementia (BPSD) increase the stress levels and coping of the partner caregiver and may lead to experiences of multiple losses for both individuals in the couple, resulting in a decrease of relationship quality [5,6,7,8,9,10,11] Conversely, couples with shared strategies for addressing challenges, and who have established strong feelings of commitment, reciprocity and couplehood, are better protected from relationship problems when living with dementia [12,13,14,15,16,17]. Further, shared activities and humour, positively influence relationship quality [18,19,20].

Research shows how music therapy and engagement in music activities support people with dementia to maintain and improve social engagement in close relationships and with their local community [21,22,23]. Shared music experiences between couples have been shown to be beneficial for quality of life and well-being, mood, and communication within home-based settings [24,25,26], institutional settings [27], and community group therapy settings [28,29,30]. Relationship quality is thematised and discussed in all these studies, though not necessarily as the primary outcome. Thurn et al. [25] studied the influence of a home-based music therapy program on relationship quality on two couples living with dementia, in relation to BPSD. They found that the music therapy intervention led to moderate improvement in relationship quality as measured by the standardized Quality of Caregiver-Patient Relationship (QCPR) scale [31], and qualitative interview analyses indicated that the music therapy intervention facilitated positive feelings and a good living environment for both included couples. The positive results from this one study, and the lack of more studies focusing specifically on relationship quality and music therapy interventions, indicates a need for more relationship-focused research within the music therapy and dementia field.

This study is a sub-study of HOMESIDE A HOME-based family caregiver-delivered music and reading Intervention for people living with Dementia: A Randomised Controlled Trial. www.homesidestudy.eu, a three-armed randomised controlled trial (RCT) comparing music therapy interventions, reading interventions, and standard care. HOMESIDE was conducted in five countries (Australia, UK, Germany, Poland, and Norway), and was funded through the EU Joint Programme on Neurodegenerative Disease Research. The primary outcome of HOMESIDE was BPSD, and one of the secondary outcome measures was relationship quality [32]. As a sub-study, the present study aims to contribute to an elaboration on this specific topic of relationship quality.

The aim of this study is therefore to find, interpret, and discuss, answers to the research question: How can a home-based music therapy intervention influence relationship quality in couples living with dementia?

The participants were couples living with dementia, while the couples were defined as two people who identify as being in significant relationship identical or equivalent to marriage. One member of the couples needed to have a dementia diagnosis. A 12-week music therapy intervention, led by a music therapist, was conducted in a home setting with the persons with dementia (PwD) and their partner caregivers (CG). Quantitative and qualitative data for measuring and gaining insight into the couples’ relationship quality were collected pre and post intervention and analysed within a mixed methods frame.

## 2. Methods

### 2.1. Study Design

This study uses some elements of the HOMESIDE research protocol, and the same music training program [26]. In HOMESIDE, dyads randomised to the music therapy intervention, in which one member of the dyad has dementia, share music experiences two to five times weekly for 12 weeks in their homes [32]. The CG within the dyad is trained and supervised by a music therapist to use music and implements these strategies to create shared music experiences with the person with dementia. The CG is also supported by fortnightly phone calls in between the training sessions. The music training program is standardised, and includes both active techniques (e.g., singing, improvisation on instruments, movement to music), receptive techniques (listening to preferred music, and the use of digital music streaming sources) and combination (e.g., movement/dancing to recorded music). In HOMESIDE, standardised assessments were administered at pre-intervention (baseline), follow-up 1 (13 weeks after baseline), and follow-up 2 (27 weeks after baseline).

The present study uses an adapted convergent mixed methods design [33]. Quantitative and qualitative data were collected at the same time-points, at baseline and post intervention (i.e., HOMESIDE follow-up 1), from two groups of participants (Table 1). One dataset (quan) was drawn from HOMESIDE participants, extracting data from couple dyads who had completed the music therapy intervention and post intervention data collection, hereafter called the HOMESIDE subset. The other datasets (quan and QUAL) were drawn from a group of four individually recruited participant couples, who completed the music therapy intervention and post intervention data collection, hereafter called individually recruited subset.

HOMESIDE subset participants received an online music therapy intervention, as the international RCT design needed to be adapted according to global COVID-19 regulations. The four individually recruited couples received the music therapy intervention in their homes, face-to-face, by their own choice. All up-to-date national guidelines on infection control were followed at all times.

The population characteristics are similar for both the HOMESIDE subset and the individually recruited subset. As qualitative data were collected from individually recruited subset only, the study design should be seen as an adapted convergent rather than a fully converged mixed methods design.

### 2.2. Inclusion Criteria and Recruitment

The inclusion criteria for this study were:Two persons living together at home, married or in an equivalent significant relationship.One of the persons had a dementia diagnosis according to the 10th revision of the International Classification of Diseases and Related Health Problems (ICD-10), as determined by a clinician experienced in diagnosing dementia.A minimum severity score of 6 on the Neuropsychiatric Inventory-Questionnaire (NPI-Q) [34], which measures the degree of behavioural and psychological symptoms of dementia (BPSD).Exclusion criteria were:Severe hearing impairment, which could not be resolved through hearing aids.No access to technical equipment or internet (HOMESIDE subset only).

These inclusion and exclusion criteria were, with exception of the final exclusion criteria, identical for both subsets, to ensure sufficient similarity between the two subsets.

The individually recruited subset consisted of four couples living with dementia. Initially, a total of 13 couples received an invitation to participate. Two of these were participants in the HOMESIDE control group, invited after completing their HOMESIDE participation, one accepted. Ten couples received the invitation through a local dementia day activities centre, two accepted. One couple was recruited through referral from a colleague, and accepted.

### 2.3. Music Therapy Intervention: Implementation of Music Therapy Activities

Over 12 weeks, all participating couples followed the HOMESIDE home-based music therapy program (Table 2). The program consisted of online meetings (for the HOMESIDE subset) or home visits (for the individually recruited subset) with a music therapist, regular phone calls from the same music therapist, and self-administered music activities in the home. The first author was the music therapist working with the individually recruited subset couples, while other music therapists from the five countries were interventionists for the HOMESIDE subset of participants. Apart from the difference between online and face-to-face intervention delivery, the same clinical protocol was followed for all participants in both subsets. The difference in delivery modality may be seen as a study limitation. However, all music therapists followed the same intervention protocol with a personally tailored program for each couple [26].

The music therapist conducted training sessions with the couples at three time points, at one, three, and six weeks after the baseline assessment and interview. Each home training session lasted approximately one hour. Between these training sessions the couple were doing music activities on their own. The therapist used conversation and music therapeutic techniques as improvisation, guided listening, singing of familiar songs, and co-creation of music with the couple, for assessing their relationship history and quality, their needs and possibilities. This gave highly individualized programs for each couple, within a person-centred and resource-oriented frame [35,36]. Four main music activities were offered and explored: singing, movement to music, playing instruments, and relaxation to music [26]. During the training sessions the music therapist and the couple or CG planned together what kinds of activities the couple should try out and use by themselves before the next training session, the planned duration and frequency of their music activities, and the music therapist advised them on useful approaches and techniques. The goals of the training sessions were to provide the couple with the resources and knowledge needed to use music in their everyday lives, and to support and guide them in their shared musical exploration. Activities that encouraged communication and positive interaction were emphasized, and the music therapist advised the CG on how the music could be used for meeting communication challenges or practical challenges. Some couples used all four activities, while others used one or two, based on their personal preferences and the assessed needs. The couple’s own musical preferences always guided the music therapist’s instructions and advice [26]. To document the couple’s use of music at home when the music therapist was not present, the couples were to record their music activities in a diary. The diary was a one-page form with simple tick-boxes and short answers, and an option to provide a richer description of the shared music experiences. The diary worked as a clinical tool, it helped the couples organise their music activities, and to remember what they had been doing from visit to visit. The participants were also asked to consent to video recording of the training sessions with the music therapist, which were used to check intervention fidelity. As the diary and video recordings were used as clinical tools and for fidelity checking, neither were part of the data analysed in the present study.

### 2.4. Data Collection and Analysis

Baseline cognitive status was assessed using the Mini-Mental State Examination (MMSE) [37]. In addition, the behavioural and psychological symptoms of dementia were assessed using the Neuropsychiatric Inventory Questionnaire, NPI-Q [34], which was part of the inclusion criteria for the whole RCT. Such data could inform interpretations of any changes in the quality of the relationship.

The Quality of Caregiver-Patient Relationship–QCPR [31] measurement scale was used for quantitative data collection at baseline and post intervention, for all included participants. The QCPR is a standardised outcome measure for relationship quality, comprising of 14 items covering emotional and relational dimensions of relationship quality. It measures the degree of warmth, conflict, and critique in the relationship (see Appendix A for the original survey). The survey is completed by the caregiving spouse in the participating couples. The QCPR has a score range of 14–70, with higher scores indicating a better relationship quality, and can be analysed by total scores and by single items.

Quantitative data were collected and managed using REDCap electronic data capture tools hosted at the University of Melbourne [38,39]. REDCap (Research Electronic Data Capture) is a secure, web-based software platform designed to support data capture for research studies, providing (1) an intuitive interface for validated data capture, (2) audit trails for tracking data manipulation and export procedures, (3) automated export procedures for seamless data downloads to common statistical packages, and (4) procedures for data integration and interoperability with external sources.

Qualitative, semi-structured interviews with the individually recruited subset couples were conducted at baseline and post intervention. Interview guides (Appendix A) were developed and were piloted with one couple not participating in the study. The interview guide for the baseline interview consisted of questions about the couple and their relationship: their common history, good days, challenges, emotional closeness, physical intimacy, and feelings toward each other. Further, the couples were asked about their musical interests. The post intervention interview focused on the experiences with music that the couple had during the intervention period, and any meaningful events they experienced as a couple in joint music-making of any sort.

The couples participated together in all interviews (except one post intervention interview in which only the CG participated, due to hospitalisation of the PwD) and are regarded as one unit consisting of two equal individuals. Both the PwD’s voice, and the voice of the CG, were heard, valued, and considered substantial as research data. The awareness of non-verbal communication, such as gestures, eye contact, touching, and other bodily expressions, can be crucial for obtaining a deeper insight of meaning in verbal conversation [14]. Such non-verbal communication was recorded as field notes and further integrated into the interview transcripts. The interviewer also paid attention to possible barriers for free communication for both individuals in the couple, such as cognitive and language challenges and relationship dynamics, and approached the interview situation pragmatically, giving time for breaks, off-topic conversations, in addition to occasional use of music as a tool for memory and joint focus. Such creative methods were used to ensure inclusion of the PwD, and to minimize the risk of misunderstandings that can cause reduced validity of the interview data [40]. All interviews were audio recorded and transcribed verbatim before analysis.

Following the analysis steps of Creswell and Plano Clark [33], the quantitative and qualitative datasets were analysed separately, before an integration process of merging the results (Figure 1). Quantitative data from the HOMESIDE subset were analysed using descriptive and inferential statistics. The QCPR scores were summarised and mean scores were identified for the total sample, both single item scores and total QCPR score. For the total scores, standard deviation was calculated, and a standard paired *t*-test was conducted for significance testing. The qualitative interviews were transcribed verbatim and included field-notes. Interview transcripts were analysed within a reflexive thematic analysis framework [41]. The NVivo software (Release 1.7) program for qualitative data analysis was used as a systemising tool for this process. In the mixed analysis and merging, there was greater weighting placed on the qualitative data, in a QUAL-quan design.

### 2.5. Ethical Considerations and Participant Involvement

Informed consent was obtained by all participants or their legal guardian. Research ethics approval was obtained for all five participating countries (see Institutional review board statement below). The trial was also registered: ACTRN12618001799246p; Clinical Trials.gov NCT03907748. This project includes persons with dementia as study participants, both through attending music training sessions with their partner and expressing their needs and preferences in the planning of music activities, and as active interviewees. The ability to give informed consent for participation for the PwDs was considered continuously, and significant changes in dementia symptoms or function were noticed and taken into account in both the clinical intervention and data collection.

## 3. Results

### 3.1. Participants

All participants were in a committed relationship (married or similar) living together in their home. Table 3 shows basic demographic details of the HOMESIDE subset and the individually recruited subset in separate columns. The individually recruited subset had solely female CGs and male PwDs. Further, there is great variance regarding cognitive function and neuropsychiatric symptoms, in both subsets.

### 3.2. Quantitative Analysis

Analysis of the 68 HOMESIDE participants’ scores on the scale QCPR showed a very small, positive change in total relationship scores from baseline to post intervention (Table 4).

While there is a large range and standard deviation, the mean and median values increase slightly from baseline to post intervention. This change is not statistically significant as tested by a standard paired t-test (*p =* 0.186). On a group level, these results indicate that there was little to no change in relationship quality for the HOMESIDE subset as measured by QCPR, during the 12-week music therapy intervention period. It should be noted that this also means that no negative change, or decrease in relationship quality, was found, and that the tendency, however weak, is positive.

The QCPR data from the individually recruited subset is displayed in Table 5. Two of the couples showed little (+2) to no (0) change in scores from baseline to post intervention. One couple showed a large positive change, total scores increasing by 9 from 56 to 65, and one showed large negative change with total scores decreasing by 11 from 66 to 55. It should be noted that in both couples who showed large change, the change of scores was evenly distributed with a +/−1 change per single items only. Thus, there seems to be no dramatic change in any single items or dimensions in any of the couples. In a small group such as this, no conclusions can be drawn from this data, other than the possible individual changes for each couple.

For QCPR, a score of ≥42 indicates a “good” relationship quality, and <42 indicates a “poor” relationship quality [31]. According to this, all the individually recruited subset couples lived in “good” relationships, both at baseline and post intervention. In the whole HOMESIDE subset, eight couples scored <42 at baseline. Four of these increased their scores to ≥42 at post intervention, moving from a “poor” to a “good” relationship. In comparison, two couples decreased their scores from ≥42 to <42, suggesting they moved from a “good” to a “poor” relationship.

The mean baseline value of the HOMESIDE subset, and the values of the individually recruited subset, are all relatively high. This may give a ceiling effect, making positive changes difficult to capture.

In summary, there is no significant indication from the quantitative data that relationship quality changed for the participant couples in this study, on a group level. A weak tendency towards an increased total score of QCPR was identified in the HOMESIDE subset. However, large standard deviation and range, indicate that the couples’ relationship quality was highly variable.

### 3.3. Qualitative Analysis

The reflexive thematic analysis revealed that the possible influences of music therapy interventions on relationship quality in the participating couples are many-faceted and complex. The life and relationship context of the couples played an important role, along with the music therapy interventions and music activities and experiences arranged by the couples themselves.

#### 3.3.1. Contextual Findings: Life, Relationship, and the Caregiver-Spouse Paradox

In the qualitative interviews, both pre and post intervention, openness and willingness to speak about their lives and relationships was evident for all couples. They all gave descriptions of loving and robust relationships, and of challenges they had met as couples throughout the years. All couples described how the introduction of dementia symptoms, and the following diagnosis, had been a turning point in their relationship. Dementia had made their lives more challenging, and their relationship roles, responsibilities, and activities had changed to varying degrees. Nonetheless, all participants stated in the baseline interviews that they saw their partner as a spouse and romantic partner, regardless of the changes that dementia had caused.

CG: Anyway, he is still my boyfriend!PwD: I agree with you (laughs)CG: Yes, we are still romantic partnersInterviewer: And [PwD], do you think of [CG] as your girlfriend?PwD: Yes. Absolutely.[Baseline interview: Couple 4, large negative change QCPR]

All the CGs expressed a feeling of a paradox between being a romantic partner and being a caregiver. They described the new caregiver role as demanding and sometimes exhausting, leading to ambiguous feelings about their roles as wives for their husbands. Three of the CGs described how feeling overwhelmed by caregiver tasks sometimes affected emotional closeness and physical intimacy in their relationship, while one couple described that keeping intimacy and sexuality in their relationship was a key to keep the feeling of being romantic partners despite demanding caregiver tasks and care recipient needs. One of the CGs who had a very strong sense of a caregiver-spouse paradox, described her feelings fluctuating between love and pain:

CG: So, I see the man that I fell in love with and have loved all these years, and still do love. He is [PwD], he can’t do all these things anymore, but he is here. So, I can hold his hand, I can help him and support him, and get gratitude in return. He shows gratitude. He becomes frustrated sometimes, and I do too, but then it’s over. I feel that we have closeness still, but of course there are things … I can’t let the pain take over, it hurts too much. The pain is there. But, I need to think positively. Take care of what we have.[Baseline interview: Couple 3, large positive change QCPR]

All the couples stated that the music therapy interventions led to using music more actively and purposefully than before, and that the training sessions with the music therapist were enjoyable and useful to them. Listening to their preferred music, sometimes with a special purpose, such as relaxing in the evening, was used most frequently and by all couples. All couples also used singing, either together or singing to each other. One of the couples made singing part of their everyday bedtime routine, while one other used singing for motivation during hikes in the forest. Three of the couples took up dancing, and one couple tried out using percussion instruments when listening to music.

#### 3.3.2. The Influences of the Music Therapy Intervention on Couple Relationship Quality

The reflexive thematic analysis resulted in a total of nine themes related to the influence the music therapy intervention had on the participating couples’ relationship quality. These themes each represent descriptions from the post intervention interviews, of ways the couples experienced having music as a more active part of their daily lives. The analysis showed that the music therapy interventions seem to have influenced relationship quality positively in many aspects. However, music may also trigger reactions in the couples that may influence their relationship negatively. There is great complexity in this, as some of the themes represent descriptions of music both as a closeness enhancer and as a trigger for feelings of distance. Figure 2 shows an organising of the themes on a continuum, where the themes with strong thematic connections are grouped into overarching themes. All nine themes are organised in either the Support of couplehood or Risk of distance overarching themes, reflecting whether the influence on relationship quality is either predominantly positive or predominantly negative. However, three of the themes are organised together in a third overarching theme additionally: Relationship vulnerabilities. This illustrates the strong connection, and potentially ambiguous value of influence, these themes may represent.

#### 3.3.3. Overarching Theme 1: Support of Couplehood

##### Sharing Memories

All four couples talked about music and memories. While participating in music together, they might be reminded of shared life experiences: their wedding, travelling, special occasions. They also used specific music to seek out certain memories and feelings that they shared and wanted to engage in. Two of the CGs used music very intentionally, looking up music that they believed would lead to specific memory recall for the PwD. All couples expressed that participating in music together helped them to hold on to their memories. Being reminded of the past, before dementia, could also in some cases lead to sadness, reminding the couple of what has been lost. In two cases couples described the music as reminding them of troubled times in their relationship history. The couples explained, however, that they tried to focus on the good memories.

CG: We reminisce … to music from our youth … I think it makes us both happy and sad.PwD: Yes, I really immerse myself in that music. And I think about everything, the way things were … the way we danced [starts crying].CG: We have so many positive memories. It’s not dangerous to cry a little(both laugh).[Post intervention interview: Couple 4, large negative change QCPR]

##### Being in the Present

While sharing memories was important to the couples, they also found it valuable to use music to stay focused in the present. When daily life was challenging, listening to music or singing could offer a new focus and help them to be present in the moment. All couples talked about the value of taking one day at the time, and living in the now, and that music can be a useful tool to facilitate this:

Interviewer: Is music a larger part of your life now?CG: Yes, it is, and I am much more aware that I can use it in different situations. Sometimes it is good to just be in the music, it gives us a rest from everything else, we can just be together in [the music] here and now.[Post intervention interview: Couple 1, small positive change QCPR]

##### Humour and Enjoyment

All four couples described having fun when being active in music, especially dancing and singing. They all told stories about how music gave them shared enjoyment, when they used music that they both enjoyed. Two couples described explicitly that music activities gave them an opportunity to take themselves and the situation less seriously for a moment, just “being silly” and laughing at themselves and each other.

CG (turned to PwD): You are more like you used to be before, when we use music. You get cheerful, I never knew you’d get cheerful from Irish music (both laugh) … and I even think you get less nervous when we sing together. That’s new.[Post intervention interview: Couple 2, no change QCPR]

##### Communication and Problem-Solving

All the couples explicitly expressed that they valued music as a means for communication, and an alternative to verbal communication. This was especially evident in two couples where the PwD had expressive aphasia, and in one couple where the PwD experienced a decline in executive functions. Two of the CGs started to use singing or music listening actively for solving practical challenges in everyday life, such as moving from one place to another, or staying calm in the dentist’s waiting room. One CG described how singing and dancing helped her husband to sit down in a chair:

(PwD had fallen asleep at this point of the interview)CG: When he is to sit down in his chair, it’s really hard for him to turn his back to the chair. And he stands like a statue, won’t move. Then one day I thought: what makes him move his legs? Yes, dancing! (laughs). So I held out my hands and asked “do you want to dance?” and started singing something like (sings a short phrase from an up-tempo song), and then he started to move his legs and turned around with me.[Post intervention interview: Couple 3, large positive change in QCPR]

##### A New Kind of Intimacy

As shown in the contextual findings, the couples regarded emotional closeness and physical intimacy as an important part of their relationship. However, physical intimacy had often changed or was in the process of changing. Three of the couples described situations in which they felt being together in music to some extent replaced or felt similar to physical intimacy. They were connected through music in a romantic or intimate way: a feeling of exclusive and unique closeness:

CG (turned to PwD) There is no one I feel as close with, as you. Sometimes, when we go to a concert, maybe together with friends, our eyes meet and we look at each other, we share memories and feelings without using words. That’s bringing us even closer, I think.[Post intervention interview: Couple 4, large negative change QCPR]

On the other hand, such intimate experiences, for two of the CGs, also reminded them of the physical and romantic partner they were in the process of losing. Such reminders were experienced as ambiguous, or “bitter-sweet” feelings.

##### Hopes for the Future

As shown earlier, living with dementia was described as a burden by the couples in this study, and the CGs often felt overwhelmed by tasks and responsibilities. This included worrying about the future: what will life be like with a partner living in a nursing home, or when becoming a widow one day? However, sharing music was described as something that gave hope for the future: that the couples imagined how being together in music could be a way to stay close and have meaningful communication even at more advanced stages of dementia.

Interviewer: Do you think music is something you can share in the future also, even though you will live in different places?CG: Yes, I think so. I think maybe our music can keep us together in the future.[Post intervention interview: Couple 1, small positive change QCPR]

#### 3.3.4. Overarching Theme 2: Risk of Distance

##### Rejection

In the interviews, three of the couples talked about experiences where the CG or PwD had taken an initiative for doing music activities, and the partner declined. In some cases, this was described as painful. Even though this theme was not as explicitly expressed and openly reflected upon as the others, non-verbal and bodily communication in the interview settings could be interpreted that the rejection was not only understood in terms of the music activity itself, but also of spending time together or even of the person him/herself. This shows that, however positive, the music activities may be perceived in the majority of incidents, they also pose a risk of inflicting pain and the feeling of being rejected.

PwD: I had hoped we could dance more.CG: But you are tired all the time, you have been tired for weeks!PwD: Yes…CG: you asked me about dancing, but there is not that much we can do with your body, you know?[Post intervention interview: Couple 4, large negative change QCPR]

##### Feelings of Inadequacy

Although the couples expressed that they enjoyed the music training sessions and activities very much, the feeling of not being “good enough” occurred from time to time, for both the CGs and the PwD. One CG described a feeling of not being creative enough, one PwD was very aware of his decline in physical function and the consequential decrease in ability to play instruments. None of the PwD, but all of the CGs, talked about their voice. Even though all couples used singing, and described it as an enjoyable and relaxed activity in general, complexes about the voice were mentioned quite frequently:

CG: Sometimes I don’t feel comfortable singing. Once you [PwD] told me I sang out of tune, that made me insecure. I don’t know, it depends on the day. Sometimes listening is better.[Post intervention interview: Couple 2, no change QCPR]

##### Reminders of Loss

Just as the music may evoke positive memories, it may also remind the couples about their losses. When remembering how things were before, through music, the couples described experiences of becoming sad or melancholic. This was not necessarily described as a solely negative experience, depending on the memory and the seriousness of the losses they were reminded of. In some cases, crying together over their losses, while listening to music, was described as enhancing emotional closeness (see the theme sharing memories). However, for two of the CGs, the music activities made dementia symptoms in the PwDs very visible to them, and they had the impression that the music in some cases increased challenging behaviours or symptoms. This caused them to sometimes avoid music:

CG: He is very rigid now, also when it comes to music. It’s because of his illness. He used to love so many types of music. Sometimes now we can’t listen to music, because he only wants one artist, and the same song over and over again. It’s too much for me when it’s like that. The music and his way of being.[Post intervention interview: Couple 1, small positive change QCPR]

#### 3.3.5. Merging of Results

The qualitative results show the complexities of relationship quality and the influence music may have on this. All the individually recruited subset couples described ways in which music and the music training sessions led to more closeness and positive, shared experiences, as well as how music at some points could trigger negative emotions or risk of feeling less connected with their partners. Some of the identified themes are also ambiguous, meaning that the musical experiences could lead to feelings of love and hurt, sometimes simultaneously. The qualitative findings are to a large degree supported by the quantitative findings, as these show no detectable change in relationship quality as measured by the QCPR. The lack of detectable change could be interpreted as an indication of stability: that the music therapy interventions may contribute to maintenance of relationship quality in couples where one could possibly expect a decline due to the dementia circumstances [6,7]. Still, as these results were not statistically significant, this finding should be interpreted cautiously.

When interpreting the merged results from this study, it appears that the music therapy interventions had an influence on relationship quality. However, a dichotomy of positive or negative influence is not sufficient, as there were no significant changes at the group level. The influences are ambiguous and complex, and determined by a multitude of individual factors in each couple. The music therapy interventions seem to predominantly enhance feelings of couplehood, such as positive emotions, closeness, and communication. However, sharing music can reveal vulnerabilities of the couple, and in some cases lead to feelings of distance.

## 4. Discussion

This study has explored the influence of a home-based music therapy intervention on relationship quality in couples living with dementia. Qualitative data analysis indicated that the music therapy intervention influenced relationship quality for the participating couples in the individually recruited subset. However, quantitative relationship quality results as measured by the QCPR scale (from both subsets) were inconclusive regarding effect. The influences identified through the qualitative analysis were predominantly positive, reflecting an enhanced feeling of couplehood for both members of the couples after participating in the music therapy intervention. Still, an ambiguity is shown, in the themes of “Risks of distance” and “Relationship vulnerabilities”. While the positive influences resemble those found in previous studies [24,25,27,28,29,30] the ambiguity and complexity found in the present study seem not to be evident in previous research. Further, it may not be possible to capture such complexity via a quantitative measurement scale, such as the QCPR. This ceiling effect was reported in two similar studies [25,29]. This ceiling effect shows that when relationship quality is the primary outcome, the QPCR score should be an inclusion criterion with baseline scores set to a level where there are moderate to severe reported relationship quality. This would eliminate ceiling effects. It is important to note, however, that the stability of the QCPR scores could indicate that relationship is sustained through a period of time in which relationship quality would probably decrease without intervention. Relationship quality for couples living with dementia has been shown to decrease over time [6], often associated with an increase of BPSD [8,10,42]. The 3-month duration of the intervention may be considered a limitation of the study. Future longitudinal studies with an intervention length of six months or more should be explored. Another limitation of the study was the lack of diversity, with only heterosexual couples living in Western, developed countries recruited and the majority of the couples in the HOMESIDE subset and all individually recruited couples consisted of a male PwD and a female CG. This lack of diversity was not intentional, but a result of practical circumstances and recruitment methods. Further studies should strive for greater diversity, regarding gender, sexual orientation and cultural background.

This study focuses on couple relationships only, with a couplehood approach [13,43]. This approach should be seen as part of a person-centred care [35], offering possibilities to maintain quality of life and health in both couples living with dementia and their partners, through focusing on the quality of their relationship. By focusing exclusively on partner relationships, this study lacks perspectives on relationship quality in other dyads or groups, which may also be substantial for quality of life and health in persons with dementia.

This study does not include any correlation analysis of cognitive level or dementia symptoms, and relationship quality. Cognitive decline and behavioural and psychological symptoms of dementia are known to have an influence on relationship quality for couples living with dementia [5], thus, the lack of such analysis could be seen as a limitation.

While previous studies show mainly positive influences on relationship quality of music therapy interventions [24,25,27,28,29,30] the identification of negative influences in the present study adds a new perspective to the subject. It shows the crucial need for professionalism and reflexivity of the music therapist in the clinical setting, both during training sessions and in follow-up contact, as the couples may need counselling and further guidance when experiencing the ambiguous influences of their shared music experiences.

## 5. Conclusions

The mixed methods analysis in this study has shown that the influence of the HOMESIDE music therapy intervention on relationship quality in couples living with dementia is complex. Quantitative results indicated stability of relationship quality, or a weak non-significant tendency towards a positive change in relationship quality, from baseline to post intervention on group level, as measured by the QCPR scale. Qualitative data analysis suggested that the HOMESIDE music therapy intervention seemed to contribute to sustaining relationship quality through enhancing positive emotions, closeness, intimacy, and communication. However, there is also a risk of evoking vulnerabilities or negative emotional responses through musical activities, thus, the support of a qualified music therapist is of great importance. Further research in this area is needed, and should strive for greater diversity, specifically regarding gender and cultural background. Longitudinal mixed methods designs that can detect both effect and complexity of relationship quality changes over time, preferably compared to a control group, would provide greater insights into the influence of music participation on relationship quality of couples living with dementia.

## Figures and Tables

**Figure 1 ijerph-20-02863-f001:**
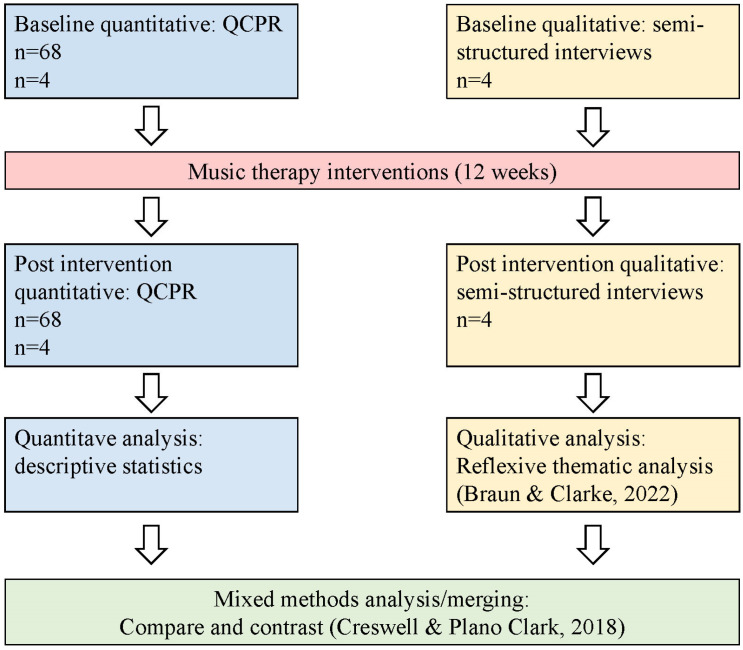
Data collection and analysis [33,41].

**Figure 2 ijerph-20-02863-f002:**
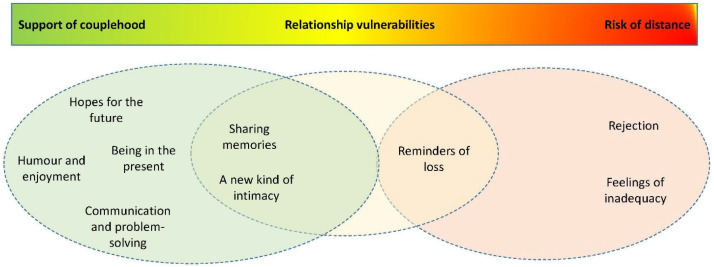
The influence of caregiver-delivered music therapy interventions on the caregiver-PwD relationship: quality themes and overarching themes.

**Table 1 ijerph-20-02863-t001:** Participant subsets.

HOMESIDE Subset	Individually Recruited Subset
n = 68 couples/136 individuals	n = 4 couples/8 individuals
Nationality: Australian (n = 19), German (n = 14), Norwegian (n = 14), Polish (n = 1), British (n = 20)	Nationality: Norwegian (n = 4)
Clinical intervention: 12-week online home-based music therapy intervention with music therapist	Clinical intervention: 12-week face-to-face home-based music therapy intervention with music therapist
Data collection: quantitative, at baseline and post intervention	Data collection: quantitative and qualitative, at baseline and post intervention

**Table 2 ijerph-20-02863-t002:** The home-based music therapy program.

Week	
1	Music training session with music therapist and the couple together
1–3	Couple experiencing music together, by themselves. Phone call from music therapist.
3	Music training session with music therapist and the couple together
3–6	Couple experiencing music together, by themselves. Phone calls from music therapist.
6	Music training session with music therapist and the couple together
7–12	Couple experiencing music together, by themselves. Fortnightly phone calls from music therapist

**Table 3 ijerph-20-02863-t003:** Demographic details.

	HOMESIDE Subset Total	Individually Recruited Participants Subset
N	n = 68 couples/136 individuals	n = 4 couples/8 individuals
Diagnosis	Alzheimer’s dementia(31), Vascular dementia,(15), Memory problems(8), Mixed dementias(4), Lewy Body dementia(3), Frontotemporal dementia(2), Semantic dementia(2), Parkinson’s dementia(1), Mild cognitive impairment(1)	Alzheimer’s dementia(2), Frontotemporal dementia(1), Parkinson’s dementia(1)
MMSE scores *	Mean 14.44 (SD 9.64) Range 29 (0–29)	Range: 4–26
NPI-Q severity scores^+^	Mean 11.03 (SD 3.58) Range 16 (6–22)	Range: 6–11
Years since diagnosis	Mean: 2.75 (SD 2.1) Median: 2.0 (IQR 3)	Range: 2–6
Years of relationship	Mean: 41.7 (SD 13.0) Median: 41.0 (IQR 18.5)	Range: 30–53
PwD age	Mean: 73.2 (SD 7.5) Median: 73 (IQR 11) Range: 34 (58–92)	Range: 58–78
CG age	Mean: 69.6 (SD 8.2) Median: 71 (IQR 11.5) Range: 39 (50–89)	Range: 55–78
PwD gender	Male: 50 (73.5%) Female: 18 (26.5%)	Male (4)
CG gender	Male: 18 (26.5%) Female: 50 (73.5%)	Female (4)
PwD education	Trade/community college(22), Bachelor’s degree(17), Master’s degree(12), Secondary/high school(11), Other(4), Doctor of philosophy(2)	Secondary/high school (2), Bachelor’s degree (1), Master’s degree (1)
PwD occupation	Professional(31), Technician(10), Manager(9), Service/sales(7), Clerical(4), Craft/trade(3), Machine operator(2), Armed forces(2)	Manager (2), Service/sales worker (1), Craft worker (1)

* 16 participants of the HOMESIDE subset had a score of 0 on MMSE, which indicates that testing was not possible. This is regarded as missing data. Thus, the number represents 75% of the subset. ^+^NPI-Q measures both severity and distress. Only severity scores are included here, in accordance with the inclusion criteria.

**Table 4 ijerph-20-02863-t004:** QCPR results for HOMESIDE subset.

	Baseline	Post Intervention	Difference	*p*-Value	95%CI
QCPR Score	Mean 54.07 (SD 8.5) Range 33 (37–70)	Mean 54.84 (SD 8.6) Range 32 (36–68)	0.765 (SD 0.1)	0.186	−1.91–0.38
	Median 54 (IQR 12)	Median 55.5 (IQR 12.5)	1.5 (IQR 0.5)		

**Table 5 ijerph-20-02863-t005:** QCPR results for individually recruited subset.

	Couple 1	Couple 2	Couple 3	Couple 4
QCPR pre intervention	50	56	56	66
QCPR post intervention	52	56	65	55
QCPR difference	2	0	9	−11

## Data Availability

The data presented in this study are available on request from the corresponding author. The data are not publicly available due to privacy and ethical restrictions.

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
