# Peer review of "The Influence of Home-Based Music Therapy Interventions on Relationship Quality in Couples Living with Dementia—An Adapted Convergent Mixed Methods Study"

_ijerph, 2023, doi:10.3390/ijerph20042863_

Round 1

Reviewer 1 Report

Through an adapted convergent mixed methods design, the goal of this study is to determine how a 12-week home-based music therapy intervention influences relationship quality in couples living with dementia. 68 participating couples from the HOMESIDE study and four individually recruited couples have received the music therapy intervention. Relationship quality for all participants has been measured by the standardised measurement scale Quality of Caregiver-Patient Relationship, and qualitative interviews have been conducted with the individually recruited participants at baseline and post-intervention. Only heterosexual couples living in Western, developed countries were included in this study. Further, the majority of the couples in the HOMESIDE subset and all the couples in the individually recruited subset consisted of a male person with dementia and a female carer. The mixed-methods analysis shows that the influence of music therapy interventions on relationship quality in couples living with dementia is complex. Quantitative results indicate a weak tendency toward a positive change in relationship quality from baseline to post-intervention on a group level, as measured by the QCPR scale. However, these results were not statistically significant. Based on the analysis of qualitative data, the music therapy interventions seemed to contribute to sustaining relationship quality through enhancing positive emotions, closeness, intimacy, and communication. However, there is also a risk of evoking vulnerabilities or negative emotional responses through musical activities; thus, the availability and professionalism of the music therapist are of great importance.

The topic of the manuscript is very relevant for the field of music therapy. Also, the manuscript is interesting on the topics of health behavior, chronic disease, and health promotion.

The cited references are mostly recent publications.

The manuscript is scientifically based, and the research plan is well thought out.

The results of the research are derived from the research plan.

The tables and figures are easy to use and show clearly what the results are when the right statistical methods are used.

The results obtained are then used to draw conclusions.

The research was conducted according to the ethical code.

The English language is appropriate and understandable.

Author Response

Thank you very much for your comments!

We are glad you found the study interesting for both music therapy and other fields, and that you argue that there is coherence between the research plan, results and conclusions.

Thank you so much for your contribution!

Sincerely,

The authors.

Reviewer 2 Report

The present article deals with the issue of home use of music therapy. I consider the article to be up-to-date and quite well prepared. The authors target the chosen topic and without unnecessary side information describe clearly stated principles of music therapy with regard to people suffering from dementia in the home environment. Although music therapy alone cannot usually serve to rehabilitate these people, the authors describe its appropriate use with a focus on the individual symtoms of people with dementia. The home environment, where many of these seniors are located, is highly appreciated. It is true that the management of dementia in the home environment is not given as much consideration. The article is professionally prepared, the data presented corresponds to the research conducted, I just did not find the research aim and problem in the text, i.e. I ask the authors to complete it. For example, this article, which discusses music therapy as a therapy that moderates certain behaviors, may be appropriate. A generalization to the conclusion can also be drawn from this article (https://www.webofscience.com/wos/woscc/full-record/WOS:000839346100001)

In the case of this addition, I then recommend the article for publication. 

Author Response

Thank you very juch for your comments!

We are glad that you found the study and manuscript mostly well-prepared and coherent.

We are also grateful for your suggestions for improvement. We will further give response to each of your points, copied from your comments, consecutively.

Sincerely,

The authors

Point 1: Extensive editing of English language and style required

Response 1: Three of the co-authors are native English speakers, and two of them have read and edited the revised manuscript. We hope this is satisfactory.

Point 2: I just did not find the research aim and problem in the text, i.e. I ask the authors to complete it.

Response 2: Aim and focus is shown first in the abstract, see line 18-20, further in the Introduction section, see line 74-76. In the Introduction section, the research question/problem is also formulated.

Point 3: For example, this article, which discusses music therapy as a therapy that moderates certain behaviors, may be appropriate. A generalization to the conclusion can also be drawn from this article (https://www.webofscience.com/wos/woscc/full-record/WOS:000839346100001)

Response 3: Thank you very much for your suggestion! We have read the article, and do not find it relevant for this paper. The target group of the suggested article is different from the one in our study, and further, this article does not take a deficit approach to intervention, but more a resource oriented approach.

Reviewer 3 Report

The manuscript addresses the important and under-covered issue of the effects of music-based intervention on the quality of the relationship in the couples living with dementia.  I was very happy to revise this paper, because I am a gerontologist with a master degree in violin and so I experienced the power of music in my life through my personal music study and in the lives of older people through my research studies.

Nevertheless, the study has got many methodological biases. So I am sorry to say that it is not ready for being published in this important scientific journal.

Introduction

In this section I really missed and I strongly suggest to add:

some epimemiological infos on the inciddence and prevalence of dementia in older population 

more information on the dementia caregiving outcomes on family caregivers of older people with dementia and especially on the couple.

Methods

The overal study design results being very confused. I really did not understand why the authors adopted differnet methods in the two sub-sample groups, this making impossible their comparability and merging the data for their interpretation. It is not understandable how QUAL data and quant data can be kept together for interpretation since they are collected by different methods and samples.

The two subset greately differ in sample size. The HOMESIDE subset includes 68 people and the individually recruited subset 8. This difference does not make the two group comparable without a statistical test.

Moreover, it is not well described if the two groups differ just in the method of recruitment or if the people belonging to the individually recruited group were trained with a different music-based programme. Please be clearer. 

Please, explain why you used also QUAL measuers for the individually recruited group and only QUANt for the HOMESIDE group.

Furthermore, the fact that the training was delivered in two different ways in the two groups is a big bias that hinder the overal study scaffold.

Among the selection criteria  is that "One of the persons have a dementia diagnosis, of any kind, or present significant memory loss/cognitive impairment"...but, Didi you use any psychometric  neurological test for screening the participants? E.g. MMSE? You mentioned the NPI-Q, but it measures the behavioural disturbances. No validated test for measuring the level of cognitive impairment?

The level of cognitive impairment also influences the capability of older people of understanding the questions of the interviews and so  the reliability of the data.

In the data collection and anlysis section, please better describe if QUANT data feed QUAL or viceversa, in other words, please specify the interaction between QUAL and QUANT data and how did you manage them for their integration. It is not clear how and where the data were mixed.

Please, clearly describe where the study was conducted: in the five countries of the larger HOMESIDE study? If so, in the analysis also the variable "country" should be taken into consideration.

Ethical consideration: please, specify if you applied to an Ethic Committee and say which one and insert the number of CLINICAL TRIAL REGISTERED. If you did not do so, please, specify why it was not mandatory.

Results

The HOMESIDE group includes people with very different level of dementia: this represents a big bias. Moreover, table 3 needs to be re-drawn by making different lines for every type of diagnosis.

The QCPR results for individually recruited subset has no statistical power, since it concers 8 subjects and so you can't compare this group with the HOMESIDE group.

Rename Figure 3 as The influence of music therapy intervention on the caregiver-PwD relationship: quality themes and overarching themes.

The QUAL analysis is very well-done and informative.

Discussion

I found this section quite weak. This mainly depends on the fact that QUAL data have been not connected to the level of cognitive impairment of the person with dementia and to the behavioral disturbances. This depending, in turn, on the lack of cognitive impairment detection at the baseline as inclusion criteria.

Very interesting are the negative outcomes of music on the relationship: this is really the main novelty taken by the study.

Minor comments

Please, pay attention to small typos along the paper e.g., line 55-56 : Thurn et al. [23] studied the influence of 55 a home-based music program on relationship quality in on two couples living with dementia.

Author Response

Thank you very much for your comments!

We appreciate your thorough feedback to our work and the manuscript. We are glad that you have a sincere interest for the field of music therapy, and for the topic of focus in particular. Thank you also for your comment on the quality of the qualitative analysis, and your notice of the novelty of this study.

We are also grateful for your suggestions for improvement. However, we do not agree with all your points. We will further give response to each of your points, copied from your comments, consecutively.

Sincerely,

The authors

Point 1: Introduction

In this section I really missed and I strongly suggest to add:

some epimemiological infos on the incidence and prevalence of dementia in older population 

Response 1: Thank you for addressing this. We have added two sentences in the very beginning of the Introduction section.

Point 2: more information on the dementia caregiving outcomes on family caregivers of older people with dementia and especially on the couple.

Response 2:  Thank you for this comment. The introduction and references 2-19 are specifically focused on family caregivers, couples and relationship quality. Any additional material that focuses on this would require cutting material from elsewhere in the article to remain in the word limit. As this information is secondary to other information, we have kept this brief.

Point 3: The overall study design results being very confused. I really did not understand why the authors adopted different methods in the two sub-sample groups, this making impossible their comparability and merging the data for their interpretation. It is not understandable how QUAL data and quant data can be kept together for interpretation since they are collected by different methods and samples.

Response 3: While using the same sample (i.e. the same participants) in both the quan and the qual sample is recommended in convergent mixed methods designs, it is not crucial (see Creswell & Plano Clark 2018, pp. 72 and 184-190). As Creswell and Plano Clark point out, using different samples could even ensure richness of perspectives, which is appreciated. However, it is not recommended to perform a complete merge of quan and qual results when the two samples are different. Consequently, we have tried to always be clear, in all sections that describe results (Abstract, Results, Discussion, Conclusion) whether we are referring to quan or qual results. The different samples, the different sample sizes (see comment below), and the uneven weighting of QUAL vs quan, is the reason why we have defined the study as an adapted convergent mixed methods design. No changes have been made to the manuscript in response to this point, and the other reviewers and the editor did not find this confusing.

Point 4: The two subsets greatly differ in sample size. The HOMESIDE subset includes 68 people and the individually recruited subset 8. This difference does not make the two group comparable without a statistical test.

Response 4: According to Creswell and Plano Clark, different sample sizes in convergent mixed methods designs is not considered a weakness. Quite the contrary, it can be a strength: “One good option is for the two samples to have different sizes, with the qualitative sample being much smaller than the quantitative sample. This helps the researcher obtain a rigorous, in-depth qualitative exploration of the phenomenon, and a rigorous, high power quantitative examination of the topic” (Creswell and Plano Clark, 2018, p. 188). We further follow Creswell and Plano Clark in their statement that when the objective of using mixed methods is combining generalizations through quan data and in-depth understandings from qual data, different sample sizes are not problematic. No changes have been made to the manuscript in response to this point.

Point 5: Moreover, it is not well described if the two groups differ just in the method of recruitment or if the people belonging to the individually recruited group were trained with a different music-based programme. Please be clearer. 

Response 5: We agree that this could be clearer, thank you for highlighting this. See our added sentence in 2.3. Music intervention: protocol and clinical methods (l. 150-154).

Point 6: Please, explain why you used also QUAL measures for the individually recruited group and only QUAN for the HOMESIDE group.

Response 6: For the HOMESIDE subset, only quan data for relationship quality was collected, as part of the RCT. The individually recruited subset was primarily qual, but quan data was also collected for the possibility to detect any important changes, similar or contrasting to the HOMSIDE subset results. See our presentation of this in 3.2. Quantitative analysis (l. 264-272).

Point 7: Furthermore, the fact that the training was delivered in two different ways in the two groups is a big bias that hinder the overall study scaffold.

Response 7: We agree that this may be a limitation of the study, however, we do not believe it represents a big bias the way you describe it. We have added a sentence about this in line 152-154

Point 8: Among the selection criteria  is that "One of the persons have a dementia diagnosis, of any kind, or present significant memory loss/cognitive impairment"… but, did you use any psychometric neurological test for screening the participants? E.g. MMSE? You mentioned the NPI-Q, but it measures the behavioural disturbances. No validated test for measuring the level of cognitive impairment?

Response 8: It is true that this was not specified, see revision in line 125-127 on diagnostic inclusion criteria. We did collect MMSE scores for both subsets, and these have been added in the demographics table. As dementia symptoms or cognitive function per se were not the focus of this study, we argue that further elaboration on this is not relevant: it was the participants’ experience of their relationship quality that was the subject of interest in this particular study. Even though MMSE scores were collected, this measure was very difficult to capture, with many participants becoming quite distressed by trying to answer the questions, there was a lot of missing data and it was not possible to correlate. However, the relationship between baseline MMSE (or equivalent) and level of relationship quality may be a factor. We highlight this as a limitation and recommend future studies include this.

Point 9: The level of cognitive impairment also influences the capability of older people of understanding the questions of the interviews and so the reliability of the data.

Response 9: The QCPR is completed by the caregiver only, and the all interviews were conducted with the couples together. See our described approach to the interviews, including specific methods for minimizing risk of reduced reliability of data in the Methods section, line 199-213. We also added one sentence under 2.5. Ethical considerations (l. 234-236)

Point 10: In the data collection and anlysis section, please better describe if QUANT data feed QUAL or viceversa, in other words, please specify the interaction between QUAL and QUANT data and how did you manage them for their integration. It is not clear how and where the data were mixed.

Response 10: As this is a convergent design, none of the strands are used to inform or “feed” the other, as would be the case in e.g. an explorative design or explanatory design. As described in 2.3. Data collection and analysis (l. 214-224) there is an uneven weighting, in a QUAL-quan design. The two strands are analysed separately, before comparing and contrasting as described in 3.3.3. Merging of results (l. 496-517).

Point 11: Please, clearly describe where the study was conducted: in the five countries of the larger HOMESIDE study? If so, in the analysis also the variable "country" should be taken into consideration.

Response 11: We agree that this could be described more clearly. We have added nationalities to Table 1.

Point 12: Ethical consideration: please, specify if you applied to an Ethics Committee and say which one and insert the number of CLINICAL TRIAL REGISTERED. If you did not do so, please, specify why it was not mandatory.

Response 12: Thank you for addressing this. The trial registration, is now added in the Ethics section (l 228-231). Ethics approvals are clearly stated under the headline Institutional Review board statement, as per the template.

Point 13: The HOMESIDE group includes people with very different levels of dementia: this represents a big bias. Moreover, table 3 needs to be re-drawn by making different lines for every type of diagnosis.

Response 13: We do agree that this represents risk of bias, and we have added this as a limitation (l. 554-557). MMSE scores at baseline have been added to be transparent about this (l 176-180, and demographics table). All participants are couples living with dementia and cognitive decline of some sort, as shown. The focus of the study is on how the music intervention may influence relationship quality, as experienced by the couples, not how cognitive decline influences relationship quality. We do, however, recognise the limitation this adds to the study.

Point 14: The QCPR results for individually recruited subset has no statistical power, since it concerns 8 subjects and so you can't compare this group with the HOMESIDE group.

Response 14: We are fully aware that these results have no statistical power, therefore, we do not include these results in any statistical analysis. However, we mean that the QCPR results of the individually recruited subset can be used to inform the findings and interpretation of mixed analysis, similarly to the results from the qualitative interviews.

Point 15: Rename Figure 3 as The influence of music therapy intervention on the caregiver-PwD relationship: quality themes and overarching themes.

Response 15: Thank you for highlighting this. We have now re-formulated the title of this figure.

Point 16: Discussion. I found this section quite weak. This mainly depends on the fact that QUAL data have been not connected to the level of cognitive impairment of the person with dementia and to the behavioral disturbances. This depending, in turn, on the lack of cognitive impairment detection at the baseline as inclusion criteria.

Response 16: We do, as earlier mentioned, agree that the lack of connection between cognitive impairment data and relationship quality data is a limitation. Still, we believe the QUAL relationship data offer valuable insights to the complex ways music interventions may influence relationship quality. This does not mean we find cognitive function or behavior irrelevant for relationship quality, conversely, we have attempted to show the importance of these factors, as it has been shown in our data, in the Results and Discussion sections. Se revisions of the Discussion section.

Point 17: Please, pay attention to small typos along the paper e.g., line 55-56 : Thurn et al. [23] studied the influence of a home-based music program on relationship quality in on two couples living with dementia.

Response 17: Thank you for highlighting this. The revised manuscript has been checked for English language errors and typos.

Round 2

Reviewer 3 Report

The Authors made a great work on the paper to address my points. Although I am not convinced about many methodological weaknesses of the study, I suggest to accept the paper for publication in its current form. Any study is perfect and the scientific community needs increased knowledge on the effects of music on dementia victims.

Author Response

Dear reviewer!

Thank you so much for your comments. We appreciate your recommendation, despite some methodological objections. 

Thank you again for your thorough review work.

Sincerely,

the authors